# A Retrospective Analysis of Feeding Practices and Complications in Patients with Critical Bronchiolitis on Non-Invasive Respiratory Support

**DOI:** 10.3390/children8050410

**Published:** 2021-05-18

**Authors:** Ariann Lenihan, Vannessa Ramos, Nichole Nemec, Joseph Lukowski, Junghyae Lee, K M. Kendall, Sidharth Mahapatra

**Affiliations:** 1Children’s Hospital and Medical Center, Omaha, NE 68114, USA; alenihan@childrensomaha.org (A.L.); varamos@childrensomaha.org (V.R.); mkendall@childrensomaha.org (K.M.K.); 2Boys Town National Research Hospital, Omaha, NE 68010, USA; nichole.nemec@boystown.org; 3Department of Neuroscience, The University of Nebraska at Omaha, 6001 Dodge St, Omaha, NE 68182, USA; jdlukowski87@gmail.com; 4Department of Biostatistics, University of Nebraska Medical Center, Omaha, NE 68198, USA; junghyae.lee@unmc.edu; 5Department of Pediatrics, University of Nebraska Medical Center, Omaha, NE 68198, USA

**Keywords:** acute respiratory failure, critical bronchiolitis, non-invasive respiratory support, early enteral nutrition

## Abstract

Limited data exist regarding feeding pediatric patients managed on non-invasive respiratory support (NRS) modes that augment oxygenation and ventilation in the setting of acute respiratory failure. We conducted a retrospective cohort study to explore the safety of feeding patients managed on NRS with acute respiratory failure secondary to bronchiolitis. Children up to two years old with critical bronchiolitis managed on continuous positive airway pressure, bilevel positive airway pressure, or RAM cannula were included. Of the 178 eligible patients, 64 were reportedly nil per os (NPO), while 114 received enteral nutrition (EN). Overall equivalent in severity of illness, younger patients populated the EN group, while the NPO group experienced a higher incidence of intubation. Duration of stay in the pediatric intensive care unit and non-invasive respiratory support were shorter in the NPO group, though intubation eliminated the former difference. Within the EN group, ninety percent had feeds initiated within 48 h and 94% reached full feeds within 7 days of NRS initiation, with an 8% complication and <1% aspiration rate. Reported complications did not result in escalation of respiratory support. Notably, a significant improvement in heart rate and respiratory rate was noted after feeds initiation. Taken together, our study supports the practice of early enteral nutrition in patients with critical bronchiolitis requiring NRS.

## 1. Introduction

Viral bronchiolitis is the most common lower respiratory tract illness and a leading cause of hospitalization of infants and young children [1,2,3]. Between 2–25% of admitted bronchiolitis patients will suffer acute respiratory failure, necessitating admission to the pediatric intensive care unit (PICU) [4,5,6]. Current mainstay of therapy emphasizes supportive care [3]. Non-invasive respiratory support (NRS) refers to the delivery of oxygen and provision of respiratory support through modalities that can allay the need for endotracheal intubation. These include continuous positive airway pressure (CPAP), bilevel positive airway pressure (BiPAP), and RAM cannula, which is a modified BiPAP delivery system that employs nasal cannula and is well tolerated by infants and young children [7]. NRS has been studied as first line treatment for acute respiratory failure and shown to be well tolerated, and has become the preferred mode of treating acute hypoxic and hypercarbic respiratory failure secondary to bronchiolitis [8,9,10,11].

Despite widespread use of enteral and parenteral nutrition in critically ill children, caloric and protein underfeeding continue to remain a common problem, including amongst the viral bronchiolitis population [12,13]. Malnutrition and nutrition deterioration are associated with longer duration of mechanical ventilation, longer PICU and hospital length-of-stay, higher risk of hospital-acquired infections, and increased mortality [14]. Previous studies have revealed NRS as a common risk factor for both delayed EN initiation and underfeeding in the PICU [15,16]. Despite data to the contrary, providers often cite safety concerns for delaying EN on NRS [17,18]. Commonly cited reasons include: (a) the potential for the patient’s status to worsen to requiring mechanical ventilation; (b) nasogastric tubes interfering with optimal NRS mask seal; (c) exacerbating respiratory failure from breaks to allow oral feeds; and (d) concerns surrounding aspiration of gastric contents during feeds [19,20]. Notably, pediatric patients often require sedation to tolerate NRS, which further raises the concern for aspiration from relaxation of airway protective reflexes [21,22]. Overall, there lacks consensus on the risk-benefit of feeding patients on NRS.

In this study, we aimed to examine the safety of enterally feeding pediatric patients with acute respiratory failure due to viral bronchiolitis managed on NRS by retrospectively examining the feeding practices within our PICU and assessing complications associated with enteral feeding. We also aimed to understand the possible benefits of early enteral nutrition on physiometric parameters and sedation needs. We hypothesized that feeding pediatric patients with critical bronchiolitis on NRS would be safe and well-tolerated. 

## 2. Materials and Methods

### 2.1. Setting 

This study was conducted at the Children’s Hospital and Medical Center, Omaha, which is the only free-standing pediatric hospital in the state of Nebraska. Affiliated with the University of Nebraska Medical Center, this 145-bed tertiary-care pediatric medical center currently houses a 32-bed combined cardiac/non-cardiac pediatric intensive care unit with an annual combined admission rate of ~1100 patients, an average daily census of 20.9, and a standardized mortality ratio of 0.87. 

### 2.2. Study Design

After obtaining approval from our institutional review board, we performed a retrospective chart review of all pediatric patients with critical bronchiolitis admitted to our PICU between January 2015 and December 2017. Informed consent was waived given the minimal risk nature of this study. This study adhered to the ethical principles outlined in the Declaration of Helsinki as amended in 2013 and was HIPAA compliant.

### 2.3. Eligibility

Eligible patients at admission were: (1) ≥37 weeks corrected gestational age, older than 72 h, and ≤2 years old, (2) carrying an ICD-9 diagnosis of acute bronchiolitis (see Appendix B), and (3) managed on NRS for acute respiratory failure, including CPAP, BiPAP, and/or RAM cannula. Exclusion criteria included: (1) either needing intubation within the first 24 h of admission to PICU or never requiring NRS during PICU stay, (2) chronic ventilator dependence, (3) immediate postoperative status, (4) single ventricle physiology, (5) active gastrointestinal bleed, (6) short-gut syndrome, (7) chronic total parenteral nutrition (TPN) dependence, and (8) any do-not-resuscitate (DNR) or other limitations in care. 

### 2.4. Variables

Demographic and clinical data collected included gender, weight, gestational age, history of prematurity, underlying neurologic or genetic abnormalities, dates of admission and discharge from the PICU, severity of illness (Pediatric Index of Mortality-III Risk of Mortality), in-hospital mortality, and presence and type of infecting pathogen. Respiratory support data collected included type and duration of NRS, incidence and duration of intubation, extubation failure, and ventilator-free-days. Nutrition data collected included mode and route of nutrition, time to initiation and to full nutrition on NRS, reported complications, and any evidence of aspiration. Data on sedation use while on NRS (type, duration, dosage) and physiometric data (heart rate and respiratory rate) prior to and after feeding initiation were also collected.

### 2.5. Definitions

Critical bronchiolitis was defined as viral bronchiolitis leading to acute respiratory failure requiring admission and management in a pediatric intensive care unit with high risk for adverse outcomes. Acute respiratory failure was diagnosed in patients requiring NRS to mitigate work of breathing and/or to keep oxygen saturation >88%. Types of NRS included in this study were CPAP, BiPAP, or RAM cannula; although heat high-flow nasal cannula (HHFNC) is a form of NRS, authors did not include this mode because most patients on HHFNC are managed on the pediatric floor at our institution and would not satisfy the definition of critical bronchiolitis. Our unit’s practice for NRS via RAM cannula is to use to a conventional mechanical ventilator to provide a set peak inspiratory pressure (PIP), peak end-expiratory pressure (PEEP), inspiratory time (i-time), respiratory rate, and fraction of inspired oxygen (FiO_2_) through specialized nasal cannulas. CPAP, defined as continuous positive airway pressure, was provided via nasal or face-mask through a conventional mechanical ventilator; BiPAP, defined as bilevel positive airway pressure, was similarly provided through nasal or face-mask through a conventional mechanical ventilator. Ventilator-free-days were defined as described previously [23]. Extubation failure was defined as re-intubation within 24 h of liberation from invasive positive pressure ventilation. Any patient receiving continuous or bolus enteral nutrition, by oral, nasogastric, nasojejunal, gastrostomy or jejunostomy tubes, while on NRS, was assigned to the EN group; patients who did not receive any feeds while on NRS were assigned to the NPO group. Optimal time to feeding initiation (within 48 h) and to full feeds (within 7 days) were defined based on the American Society for Parenteral and Enteral Nutrition (ASPEN) guidelines [14]. Complications after feeding initiation were recorded and are detailed in Appendix A. Aspiration was defined after feeding initiation as any documented feeds found in the nasopharynx with subsequent increased work of breathing. Physiometric parameters refer to heart rate and respiratory rate that were recorded for the EN group prior to and after feeds initiation. 

### 2.6. Statistical Analyses

For two group comparisons between the NPO and EN groups, the Wilcoxon rank-sum test (in case of normal distribution failure) or the chi-square test of independence was used. Continuous variables are presented with medians and interquartile ranges, while categorical variables are described using frequencies and percentages. Demographic and clinical data relating to age, NRS duration, PICU length of stay, pathogen burden, intubation duration, and ventilator-free-days were compared between groups using the Wilcoxon rank-sum test. All other categorical data were compared using the chi-square test of independence. Nutrition data for the EN group is presented as frequencies and percentages. Physiometric parameters (heart rate and respiratory rate) as continuous variables were tested for normality and analyzed using paired Student’s *t*-test. Sedation use between groups was compared using the Mann–Whitney U test. Statistical significance was established two-sided at *p* < 0.05. 

## 3. Results

### 3.1. Eligibility

We identified 342 pediatric patients admitted to our pediatric intensive care unit between January 2015 and December 2017 with a diagnosis of critical bronchiolitis. After excluding 164 patients (for never requiring NRS (75), requiring intubation within the first 24 h (65), having chronic ventilator dependence (12), falling out of the age criteria (6), having single-ventricle physiology (4), or being fresh post-operative status (2)), 178 eligible patients with critical bronchiolitis managed on NRS were stratified into NPO (*n* = 64, 36%) and EN (*n* = 114, 64%) groups based on their feeding status during their stay (Figure 1).

### 3.2. Patient Demographics

When comparing demographic and clinical data between the two groups, there were no significant differences in gender, past medical history of prematurity, underlying genetic or neurologic abnormalities, severity of illness, or overall mortality. However, median age in the EN group was significantly lower than that of the NPO group (3 months, IQR 2–6 vs. 10 months, IQR 3–16; *p* < 0.001) with a higher proportion of infants in the former group (93% vs. 59%; *p* < 0.001). Not surprisingly, median weight was also lower in the EN group (5.7 kg, IQR 4.7–7.4 vs. 8.4 kg, IQR 5.4–10.5; *p* < 0.001) (Table 1). 

### 3.3. Clinical Characteristics

When examining ventilation and pathogen characteristics for all patients, groups were similar with respect to the type of NRS (RAM vs. CPAP vs. BiPAP), pathogen number and type. More specifically, a majority of patients in both groups were managed on RAM cannula (≥95%); had a single pathogen etiology (>70%) causing critical bronchiolitis, mostly viral in origin (>90%) (Table 2 and Appendix A). However, the NPO group had both shorter median NRS duration (1 day, IQR 0.8–2 vs. 3 days, IQR 2–4; *p* < 0.001) and overall PICU length of stay compared to the EN group (2 days, IQR 1–3 vs. 3 days, IQR 2–5; *p* < 0.001) (Table 2); the latter difference was lost upon intubation (11 days, IQR 6–32.5 vs. 15 days, IQR 12.5–25.5; *p* = 0.38, Appendix A). Moreover, while intubation rates were generally low in the entire cohort, the NPO group had a 3.5-fold higher incidence of intubation compared to the EN group (14% vs. 4%; *p* = 0.016) (Table 2). Within this cohort of intubated patients, characterized by a high incidence of multi-microbial infections (median 3 pathogens per patient), the EN group had a significantly higher incidence of bacterial superinfection compared to the NPO group (100% vs. 45%, *p* = 0.038), while the NPO group had a significantly shorter median NRS duration prior to intubation (1.5, IQR 1.1–2.3 vs. 2.5, IQR 1.9–4.5; *p* = 0.044). No statistically significant differences were discerned with respect to median weight, age, time to and duration of intubation, extubation failure, ventilator-free-days, or severity of illness (Appendix A).

### 3.4. Enteral Nutrition Details 

Amongst the non-intubated EN group of patients (*n* = 109), we delved deeper to dissect feeding practices and complications (Table 3). Overall, the mode and route of highest physician preference was continuous (78%) via the nasogastric route (63%). Ninety percent of patients in this group had feeds initiated within 48 h of NRS initiation (which corresponded closely with PICU admission) and 94% reached full caloric goal within 7 days. Median time to initiation of enteral feeds was 19 h (IQR 11–37) and median time to full feeds was 40 h (IQR 24–58) after NRS initiation. Complications with enteral nutrition were encountered in only 8% of patients with only one patient having documented evidence of aspiration (Appendix A). Of note, none of these patients experienced any escalation in NRS support.

### 3.5. Physiometric Parameters and Enteral Nutrition

Finally, we examined the effect of initiating feeds on physiometric parameters, i.e., heart rate and respiratory rate. Within the same cohort of non-intubated EN patients (*n* = 109), we found a significant decrease in both heart and respiratory rate after feeds were initiated (Figure 2). More specifically, average heart rate declined from 140 to 129 beats per min post-feeds initiation (*p* < 0.001), while average respiratory rate declined from 52 to 45 breaths per min post-feeds initiation (*p* < 0.001). Of note, average sedation needs (for dexmedetomidine and lorazepam) where not different between NPO and EN groups (Appendix A).

## 4. Discussion

In this retrospective single-center study over a 36-month period, we reviewed feeding practices of 178 patients with critical bronchiolitis requiring NRS. We found that in our institution, a majority of these patients (64%) were started on enteral nutrition, most within 48 h of initiation of NRS, reaching full feeds in less than a week. The overall complication rate in the EN group was relatively low with a <1% incidence of aspiration. However, none of these complications resulted in escalation of respiratory support or worsening respiratory failure. Our practices closely align with current American Society for Parenteral and Enteral Nutrition (ASPEN) [14] and the European Society of Pediatric and Neonatal Intensive Care (ESPNIC) [24] recommendations to initiate feeds for critically ill patients within 24–48 h of admission to the PICU and achieve up to two-thirds of goal feeds within 7 days. On the other hand, they contrast with a prior pediatric study associating non-invasive support with delays in feeds initiation [15]. Our findings also diverge from a large French observational adult study that reported increased morality, increased invasive mechanical ventilation needs, and shorter ventilator-free-days in critically ill patients fed enterally [25]. In addition to suggesting overall safety of early enteral nutrition while on NRS, physiometric data in this group associated benefit to feeding as evidenced by both lower heart rate and respiratory rate, despite no change to overall sedation needs. Though causality cannot be discerned at this time, the noted trends imply improved patient comfort with feeds alone. 

We found that many demographic and clinical parameters between groups remained similar, including gender, prior history of prematurity, neurologic or genetic abnormalities, mode of NRS, pathogen number and type, illness severity, and overall mortality. That said, a key difference between groups was age (and weight), with patients in the EN group being considerably younger with a median age of three months (and in turn smaller). Could this have contributed to increased provider concern regarding the risk of caloric and protein deficits in these patients, and in turn, a higher likelihood to initiate enteral nutrition? A prior multi-center retrospective study of over 5000 critically-ill children reported similar observations of younger patients having early enteral nutrition, especially if they were less severely ill [26]. Similarly, while we were surprised with the longer duration of NRS, and consequently longer PICU length of stay, in this group, a plausible explanation might rest in provider preference to wean NRS slower in younger patients with acute respiratory failure. Notably, in contrast to adult studies [25,27], we showed no difference in mortality or ventilator-free-days between groups and reduced intubation rates in the EN group, despite similar severity of illness and a higher incidence of bacterial superinfections compared to the NPO group. These interesting observations remain distinguishing features of the EN group that warrant further investigation. 

The NPO group experienced shorter NRS duration, a higher incidence of intubation, and shorter PICU length of stay. We perceive these differences to highlight two distinct populations of patients, i.e., one relatively healthy with rapid recovery and one sicker cohort. The intubated NPO patients with a shorter duration of preceding NRS would constitute the latter, albeit smaller group, i.e., these patients were sicker at the outset. At present, we cannot determine if NPO status contributed to worsening respiratory failure in these patients secondary to agitation or caloric deficits or if patients were kept NPO appropriately due to concern for declining status. The second “healthier” group remained not only unintubated but also had a shorter duration on NRS and thus a shorter PICU length of stay (by ~1.5 days) compared to EN group. Given the relatively older demographic of this group, this observation would not be surprising. The example of an overall healthy 10-month old female with acute respiratory failure due to bronchiolitis spending the median 48 h in the PICU, recovering and having feeds initiated is not far-fetched. Moreover, providers might argue that an older, and presumably larger, child might tolerate NPO status better than a younger one. 

Pediatric studies have shown benefits of early nutrition, in both mechanically ventilated and non-invasively supported patients [16,28]. In fact, a recently published retrospective study in four European PICUs reached similar conclusions to ours, reporting safety and tolerance of early enteral nutrition in patients managed on NRS [20]. However, we now need larger scale prospective studies specifically designed to examine the effects of early enteral nutrition on pediatric intubation rates, ventilator-free-days, organ failure-free-days, and overall length of PICU and hospital stay. Similar to our work, these studies should further dive into the influence of EN on physiologic parameters as a proxy for patient tolerance and comfort and the overall disease trajectory. 

With new emphasis on the benefits of early enteral nutrition spotlighted by national societies like ASPEN and ESPNIC, the trend towards this practice shown here is encouraging. Despite the single-center retrospective nature of our study, we have added valuable insight on the moving practice of early enteral nutrition in critically-ill pediatric patients. We feel confident purporting the safety and possible benefits of feeding patients suffering from respiratory failure requiring non-invasive respiratory support. 

## Figures and Tables

**Figure 1 children-08-00410-f001:**
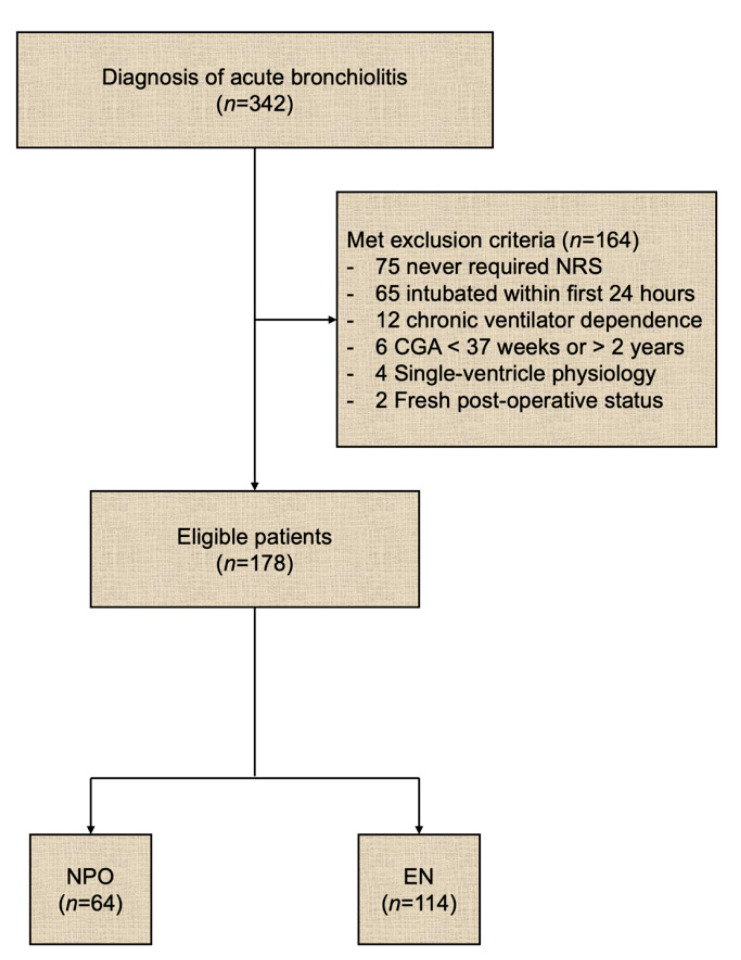
Flow diagram of patient enrollment. CGA = corrected gestational age, EN = enteral nutrition, NPO = nil per os (nothing by mouth), NRS = non-invasive respiratory support.

**Figure 2 children-08-00410-f002:**
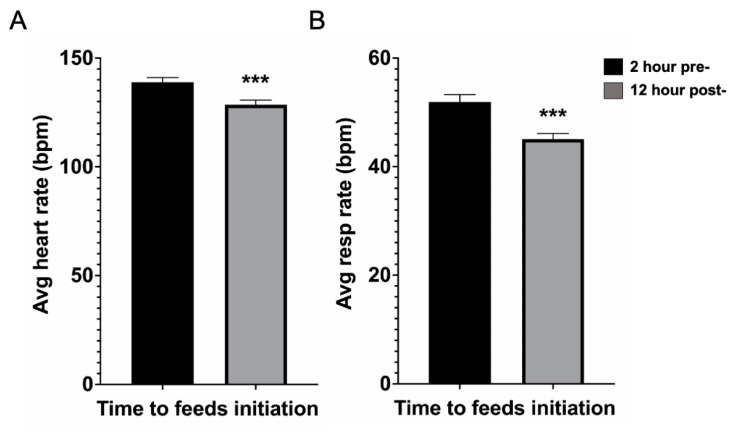
Physiometric parameters for the enteral nutrition (EN) group demonstrating (**A**) heart rate and (**B**) respiratory rate trends prior to (2 h pre-) and after (12 h post-) EN initiation. Data presented as mean ± standard error of mean. *** *p* < 0.001 by paired two-tailed Student’s *t*-test. Avg = average, bpm = beats/breaths per minute, resp = respiratory.

**Table 1 children-08-00410-t001:** Demographic Data for Entire Cohort (*n* = 178).

	NPO (64)	EN (114)	*p*
Sex, *n* (%)			
Male	39 (61%)	70 (61%)	1.0
Weight (kg), median (IQR)	8.4 (5.4–10.5)	5.7 (4.7–7.4)	<0.001
Age (months), median (IQR)	10 (3–16)	3 (2–6)	<0.001
Age, *n* (%)			
≤1 month	7 (11)	13 (11)	<0.001
2–12 months	31 (48)	94 (82)	
13–24 months	26 (41)	7 (6)	
Maturity, *n* (%)			
Full-term at birth	41 (64)	68 (60)	0.56
Pre-term at birth	23 (36)	46 (40)	
Late (33 to <37 weeks)	15 (23)	33 (29)	0.79
Very (28 to <32 weeks)	5 (8)	7 (6)	
Extreme (<28 weeks)	3 (5)	6 (5)	
Genetic abnormalities, *n* (%)	12 (19)	12 (11)	0.12
Neurology abnormalities, *n* (%)	7 (11)	9 (8)	0.50
PRISM-III ROM, % (IQR)	0.5 (0.5–0.8)	0.5 (0.3–0.6)	0.57
Mortality, *n* (%)	0 (0)	0 (0)	-

IQR = interquartile range, PRISM-III ROM= risk of mortality based on Pediatric Risk of Mortality-III score. Wilcoxon rank-sum test used to compare continuous variables presented as median (interquartile range); Chi-square test of independence used to compare categorical variables presented as *n* (%).

**Table 2 children-08-00410-t002:** Clinical Characteristics of Entire Cohort (*n* = 178).

	NPO (64)	EN (114)	*p*
NRS duration (days), median (IQR)	1 (0.8–2)	3 (2–4)	<0.001
Type of NRS, *n* (%)			
RAM	52 (95)	108 (99)	0.34
CPAP	0 (0)	1 (1)	
BiPAP	3 (5)	2 (2)	
Intubation ^#^, *n* (%)	9 (14)	5 (4)	0.016
Pathogen *^f^* (#), median (IQR)	1 (1–2)	1 (1–1)	0.21
Pathogen positive, *n* (%)	57 (89)	105 (92)	0.5
Single	41 (72)	81 (77)	0.53
Multiple	16 (28)	25 (24)	
Virus only	53 (93)	99 (94)	0.74
Virus + Bacteria	4 (7)	6 (6)	
PICU LOS (days), median (IQR)	2 (1–3)	3 (2–5)	<0.001

^#^ Refer to Appendix A for details; *^f^* refer to Appendix A for details; NRS = non-invasive respiratory support, IQR = interquartile range, CPAP = continuous positive airway pressure, BiPAP = bilevel positive airway pressure, PICU LOS = pediatric intensive care unit length of stay, Wilcoxon rank-sum test used to compare continuous variables presented as median (interquartile range); Chi-square test of independence used to compare categorical variables presented as *n* (%).

**Table 3 children-08-00410-t003:** Nutrition Details for Non-Intubated EN Group (*n* = 109).

	EN (109)
Route, *n* (%)	
PO	11 (10)
NG	69 (63)
NJ	14 (13)
GT/JT	15 (14)
Mode, *n* (%)	
Bolus	24 (22)
Continuous	85 (78)
Time to initiation (h) *, median (IQR)	19 (11–37)
*n* (%)	
≤48 h	98 (90)
>48 h	11 (10)
Reached full EN (h) *, median (IQR)	40 (24–58)
*n* (%)	
≤7 days	103 (94)
>7 days	6 (6)
Complications ^#^, *n* (%)	
Yes	9 (8)
No	100 (92)
Evidence of aspiration, *n* (%)	
Yes	1 (1)
No	108 (99)

* Hours after NRS; ^#^ refer to Appendix A for details; PO = per os (by mouth), NG = nasogastric, NJ = nasojejunal, GT/JT = gastrostomy tube/jejunostomy tube, EN = enteral nutrition, h = hours.

## Data Availability

Data available upon request.

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
