# Peer review of "A Retrospective Analysis of Feeding Practices and Complications in Patients with Critical Bronchiolitis on Non-Invasive Respiratory Support"

_children, 2021, doi:10.3390/children8050410_

Round 1

Reviewer 1 Report

The manuscript is well written and the the project as high scientific interest. The manuscript states how the  feeding (EN) of pediatric patients on NRS is not only safe but ameliorate the clinical condition related to specific parameters. Nevertheless, there are some minor aspects that could make more clear the study and improve its quality.

1) In my opinion in the Abstract I would briefly explain the meaning of the acronyms as (EN) and NPO since this become clear in the Material and Methods

2) In my opinion the Authors did not clearly explained the safety concerns that allow the delay of EN on NRS in the Introduction and the point a), b) and c) and they do not specify the references and the data.

3) In general the table in supplemental material are not really easy- to-read and to follow. I would suggest to add a column to separately put the absolute value of the data (n) and the percentage of the related absolute value if this may improve the reading of them.

4) Line 171 I would add “no statistically significant differences” since there are differences.

5) In Figure 2 The x-axis of bar graph is not clear and the font is too big. I would rather put “time of feed initiation” and a legend to discriminate between the black bar (2 hour pre feeding) and the grey one (12 hours post-initiation).

6) In the same graph of figure 2 the statistic in my opinion is not convincing. Even if the samples are a large number, I would not expect the strong statistic (p<0.001) with the comparison between the two bars that differ only for 11 bpm of the heart rate and 7 respiratory rate (bar graph 2). To this purpose, I strongly suggest to either reproduce the statistic with the Wilcoxon non parametric test admitting that the samples are dependent since they are the heart beat value of the same patients in 2 different condition or, I would suggest to perform the parametric t-test (two-tailed) since in this case you might want to consider how the mean value of a particular distribution could deviate significantly from a certain reference value. If my request does not fit with these data I would ask the Author to argue this choice and to provide the original data related to the patients.

7) Line 219. I would indicate the meaning of ASPEN and ESPNIC

8) I would add details in Enteral Nutrition Details (e.g bolus quantity) 

Reviewer 2 Report

This study was a retrospective review (n=178) looking at enterally feeding pediatric patients on non-invasive respiratory support due to acute respiratory failure secondary to bronchiolitis. This was an interesting and needed study, considering the lack of evidence on respiratory support and feeds in general. Out of 178 children, 36% were NPO, while the balance had enteral feeds (EN). Compared to the EN group, the NPO group had significantly higher weight, were older, had shorter PICU LOS and duration of NRS (when not intubated), higher rates of intubation (though incidence was low, n=14), more multi-microbial infections (for the intubated subgroup). In the non-intubated EN group, the majority were NG, continuous, initiated within 48 hours, and reached TPN in less than 7 days. Complications to feeds were low at 8%, with only 1 out of the 109, reporting aspiration. Heart rate and respiratory rate after feeds were significantly lower post feeding.

Introduction

  • Line 42/43. Please provide a reference for “…and RAM cannula, which is a modified BiPAP delivery system that employs nasal cannula and is well tolerated by children under 2 years of age.”
  • The aim of the study appeared to be both a description of the population and safety. It would be helpful if authors could provide the hypothesis they had prior to starting the study.

Methods

  • State the disease severity score was used - PIM-III ROM = Pediatric Index of Mortality-III Risk of Mortality
  • Line 88: suggest changing “extubation success” to “extubation failure”, since that is what is defined later.
  • Line 103 …a Servo ventilator (not ventilators). Related to this, was this the brand “Servo”? If yes, provide company details (and even better the exact name of the ventilator).
  • For non-skewed continuous data, was a t-test considered (rather than Mann-Whitney U)?

Discussion

  • Authors note the differences in age and duration of NRS and PICU LOS in the EN group compared to the NPO group (starting on line 232). Indeed, I found these results interesting as well. Although the literature on children receiving or not receiving feeds in the PICU may be lacking (especially for bronchiolitis), it would be helpful for authors to provide descriptions from prior references on feeding practices for PICU patients and characteristics of NRS in the PICU, especially differences in age. This will support the speculations provided by the authors (and help readers to seek out potential answers).
  • Same comment for the paragraph describing intubation, and health status of the infants. Although the intubation group was small, it would have been helpful to provide a description (PIM-III ROM in supplementary table) The speculations provided were great, but it would be helpful to also supplement them with related evidence from prior studies.
  • Comments on the weight difference? Or integrate with comments of age.
  • Line 261: define VFD
  • Last paragraph provided some guidance with respect to future trials. Please include aspects of your own study results. For example, future trials should consider age/weight, prognostic factors, measure patient comfort directly.
  • In addition, it would be interesting to see a comparison of HR and RR with the NPO group for approximately the same that feeds were provided to EN group. Is this a possibility? If not, please suggest this as a next step.

Minor comments

  • Define non-invasive respiratory support as NRS then use throughout (no need to put full term later). Same comment as “nil per os” (NPO).
